# Analyzing Meta-Heuristic Algorithms for Task Scheduling in a Fog-Based IoT Application

Dadmehr Rahbari 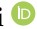

Communication System Research Group, Thomas Johann Seebeck, Department of Electronics, School of Information Technologies, Tallinn University of Technology, 19086 Tallinn, Estonia; dadmehr.rahbari@taltech.ee

**Abstract:** In recent years, the increasing use of the Internet of Things (IoT) has generated excessive amounts of data. It is difficult to manage and control the volume of data used in cloud computing, and since cloud computing has problems with latency, lack of mobility, and location knowledge, it is not suitable for IoT applications such as healthcare or vehicle systems. To overcome these problems, fog computing (FC) has been used; it consists of a set of fog devices (FDs) with heterogeneous and distributed resources that are located between the user layer and the cloud on the edge of the network. An application in FC is divided into several modules. The allocation of processing elements (PEs) to modules is a scheduling problem. In this paper, some heuristic and meta-heuristic algorithms are analyzed, and a Hyper-Heuristic Scheduling (HHS) algorithm is presented to find the best allocation with respect to low latency and energy consumption. HHS allocates PEs to modules by low-level heuristics in the training and testing phases of the input workflow. Based on simulation results and comparison of HHS with traditional, heuristic, and meta-heuristic algorithms, the proposed method has improvements in energy consumption, total execution cost, latency, and total execution time.

**Keywords:** fog computing; task scheduling; meta-heuristic algorithm

## 1. Introduction

Extensive advances in Internet of Things (IoT) applications on the one hand, and the advent of next-generation networks, such as 5G and beyond, have made public and specialized interest in areas such as computing and telecommunications more evident. IoT includes every online object, such as smart cameras, wearable sensors, environmental sensors, smart home appliances, and vehicles [1].

Currently, the number of connected devices is greater than the number of people on earth, and this number is increasing every day without any limitations. The IoT increases the quality of human life, but use of the IoT produces massive amounts of data, which creates an excessive burden for data storage systems and analysis [2].

Some categorized computing paradigms such as edge computing, mobile edge computing (MEC), cloud computing, mobile cloud computing (MCC), and FC are as follows.

- Edge computing enables data processing at the network edge. It provides fast responses to computational service requests. Additionally, it does not associate IaaS, PaaS, SaaS, and other cloud-based services spontaneously and concentrates more on the end-device side.
- MEC is an evolution of cellular base stations. It can be connected or not connected to distant cloud data centers. MEC uses radio network information in distributed applications [3].
- Cloud computing is used to manage and control the massive amount of data produced by objects. Many applications, such as health monitoring, intelligent traffic control, and games, may need to get feedback in a short amount of time, and the latency caused by sending data to the cloud and then returning the response from the cloud

to the operator of these programs has adverse effects. Further, the massive amount of data generated by some of these applications may impose heavy burdens on the network. Sending this volume of data to the cloud and then returning it is not desirable [4]. Cloud data centers are centralized, so it is difficult to service distributed applications. Using cloud computing for these applications increases latency and network congestion and decreases quality of service (QoS) [3].

- MCC provides necessary computational resources to support remote execution of offloaded mobile applications in closer proximity to end-users based on a three-tier hierarchical architecture. MCC combines cloud computing and mobile computing [3].
- Fog computing is a type of distributed computing and is located between objects and the cloud. FC extends clouds to the edge of the network and presents a solution to overcome its limitations. FC can also provide MEC, MCC, and edge computing [5].

FC can spread on a single node or several nodes. Increasing the number of FDs increases the scalability, flexibility, and computing power of the system. Tasks of FDs include analyzing, filtering, and temporary storage of data. FC combines the advantages of cloud computing and edge computing. FC includes benefits such as location awareness, mobility support, real-time interactions, scalability, and interoperability. Some other FC advantages are found in [6]. Reducing this physical distance reduces latency and provides real-time responsiveness. Initial operations such as pre-processing, filtering, and compressing are performed on raw data; thus, a small amount of data needs to be sent to the cloud, which, in turn, reduces network traffic. FC increases the sleep time of devices and decreases the energy consumption of sensor devices.

Cloud computing and FC have common characteristics; however, FC has more features than cloud computing, including geographical distribution, real-time interaction, support mobility, heterogeneity, and interoperability [2]. When computing power is needed, multiple FDs can be implemented instead of a single FD for computation, which increases scalability and flexibility. FC has some challenges. These challenges include infrastructure compatibility (interoperability, monitoring, and responsibility), virtualization (resource lifecycle and context awareness), scheduling and resource management (location of resources, task scheduling, and offloading), programmability (usability and session management), and security. Scheduling of resources and security are particularly important.

Scheduling in distributed environments is generally divided into three categories: resource scheduling, workflow scheduling, and task scheduling [7]. Scheduling is defined as follows: finding an optimal solution for allocation of a set of resources $R = \{R_1, R_2, ..., R_m\}$ to a set of tasks $T = \{T_1, T_2, ..., T_n\}$ or a workflow. Scheduling can be done by deploying a set of predefined constraints and objective functions [8]. A task is a small part of the work that must be performed within a specified time. One of the goals of task scheduling is to maximize the use of existing resources and minimize the waiting time of jobs [9]. Service providers and service users are the other scheduling beneficiaries. Service user interest corresponds to makespan, budget, deadline, security, and cost. On the other hand, the service provider's objective is load balancing, resource utilization, and energy efficiency. The use of different objectives for scheduling in various research has included: makespan (45%), cost (16%), deadlines (16%), load balancing (16%), and budget (7%) [10]. Makespan is the end time of the last job. The aim is to minimize the makespan. In recent years, task scheduling has been widely used in distributed computing systems.

There is a difference between task scheduling in Cloud and FC environments. In the cloud, the resources are placed in different real or virtual machines (VMs). Thus, the scheduling strategy must search a pool of resources to allocate the best of them to the requested task. In FC, there is a hierarchical architecture, which means different layers of FDs, and there is also a cloud data center on top of these. Thus, the scheduling method first searches for the best resources in the fog layer. If there are no suitable resources, then the cloud can be searched for resources.

In the following, there are some terminologies such as heuristic, meta-heuristic, and hybrid heuristic. The Heuristic method is a search in space of solutions to find the optimal

result. Meta-heuristic algorithms can be used for selecting and producing hard computational problems (NP-hard). Meta-heuristics are based on finding the best heuristic among several heuristics. Hybrid heuristics is the combination of two or more meta-heuristics to make use of their advantages. Hyper-heuristics is the selection of one method out of several meta-heuristic algorithms.

This work is an extension of [11], which presented the HHS algorithm for allocating suitable resources to application modules in FC. The main contributions of this paper to the base paper can be expressed as being more surveying of the literature, technical explanation, analysis, and evaluation. In fact, a hyper-heuristic algorithm based on some low-level heuristic methods such as genetic algorithm (GA), particle swarm optimization (PSO), ant colony optimization (ACO), and simulated annealing (SA) are compared with traditional and heuristic methods (First Come First Served (FCFS), concurrent, and Delay-Priority (DP)) [12] and meta-heuristic methods such as Non-dominated Sorting Genetic Algorithm, third version (NSGA-III) [13] and Many-Objective PSO (MaOPSO) [14]. In addition to the former metrics, e.g., energy consumption, cost, total network usage, network lifetime, and latency, another important parameter—the number of users—is taken into account.

The rest of the paper is organized as follows. In Section 2, related works are presented. The proposed HHS scheduling algorithm is presented in Section 3. In Section 4, the experimental results of the simulation are provided. Section 5 presents conclusions and future work.

## 2. Related Work

FC has many studies in the field of resource scheduling. Deng et al. [15] provided a scheduling methodology for the cloud–fog environment. Their scheduler is based on the efficiency of energy consumption and the reduction of transmission delays. The simulation results in MATLAB demonstrate a reduction of bandwidth consumption by resources and a decrease in latency. Intharawijitr et al. [16] proposed a scheduling method to reduce the amount of computation and latency in FC. Their scheduling policy for selecting the best FDs is random, least latency, and the most remaining resources. Their simulation results indicate that the second policy of least latency is more effective in improving system performance. Scheduling algorithms in the cloud or at edges are divided into two categories: (1) traditional or classical algorithms (based on law and suitability for small scheduling problems) and (2) intelligent algorithms [8].

### 2.1. Traditional Algorithms

In a FCFS scheduling algorithm, when a new task arrives, it is placed at the end of the queue. The first task at the beginning of the queue always runs first. This method has easy implementation. The round-robin (RR) method is based on the FCFS method for scheduling tasks so that resources are allocated to tasks for fixed periods. The advantage of this approach is load-balancing [7].

In a Min–Min scheduling algorithm, the smallest job out of available tasks is selected; then, it takes a resource the minimum possible time to finish. This method increases makespan. The Max–Min method selects the longest work among existing tasks and assigns it to the fastest machines to run. In this method, smaller tasks must wait for a longer period, and this increases the makespan. However, this method has a better makespan than its peers' algorithms [9].

In a priority scheduling algorithm, tasks are classified based on their priority. These priorities are considered using QoS parameters. Then, resources with the best completion time are assigned to these tasks [17].

In [12], FCFS, concurrent, and DP scheduling methods are used for task scheduling in the iFogsim environment. They provided two case studies, EEGTBG and VSOT, then analyzed the results by delay, total network usage, and the number of application modules based on the number of users. The results show that minimum delay is obtained by the DP method, and minimum network usage occurs with the concurrent method.

## 2.2. Heuristic Algorithms

The security-aware and budget-aware (SABA) algorithm [18] is defined for scheduling in a multi-cloud. SABA contains three main steps: clustering, prioritizing tasks, and assigning data to specific data centers based on the constant workflow datasets. SABA's objectives are makespan, security, and budget.

The Multi-objective Heterogeneous Earliest Finish Time (MOHEFT) [19] scheduling algorithm is known as a model of HEFT. This heuristic algorithm is based on Pareto solutions. Makespan and cost optimization are based on workflow applications on the Amazon commercial cloud. The flexibility provided by MOHEFT as a multi-objective algorithm is very attractive. Tests have shown that in some cases, costs can be reduced to half by a small increase of 5% in makespan.

The enhanced IC-PCP with replication (EIPR) algorithm [20] is a scheduling and provisioning solution that uses the idle time of VMs and a budget surplus to repeat the tasks to meet the deadline. This method improves the performance of the network. Additionally, in [21], tasks are scheduled by a heuristic algorithm such that the objective function includes the makespan and the execution cost of the tasks. The results show higher efficiency and lower mandatory costs than other methods.

## 2.3. Meta-Heuristic Algorithms

### 2.3.1. GA-Based Meta-Heuristic Algorithms

GA was defined in 1975 by Holland [22]. Some of the schemes schedule workflow using the original GA, and the others produce a better initial population [23] to achieve better results. Szabo et al. [24] have proposed two chromosomes in GA for resource allocation problems. One chromosome is responsible for assigning nodes, and the other is responsible for ordering. The results show that the method runtime improves between 10 and 80 percent during data transmission.

Wang et al. [23] used a multi-objective GA to optimize the energy consumption and increase profits for the service provider. The Pareto principle is used for the optimal choice between available solutions based on current needs. Simulation results in CloudSim show that energy consumption is reduced by 44.46% and 5.73% rates for the service provider.

The authors of [13] presented the NSGA-III. This is a many-objective optimization method that can find a well-distributed set of trade-off solutions where a few preferred reference points are supplied. In [25], resources are allocated to tasks in FC by the NSGA-II method. This work is simulated in MATLAB. They only compare their method with the random allocation method. Their scheduling method reduces the latency and improves the stability of task execution.

### 2.3.2. ACO-Based Meta-Heuristic Algorithms

ACO was introduced by Marco Dorigo and was inspired by the behavior of some species of ants. Ants guide each other by pouring pheromones on the ground. This behavior is used for solving optimization problems [26]. ACO as a meta-heuristic algorithm is used for many optimization problems that include scheduling. Liu et al. [27] used the ACO method to deposit pheromones between VMs in order to achieve the past utility of placing pheromones in physical machines. Their algorithm is simulated in a homogeneous environment, and only CPU and memory resources are considered. Tawfeek et al. [28] present task scheduling based on ACO. The main objective of this algorithm is to minimize makespan. Random ACO search engine optimization is a method that allocates VMs to entry tasks. The simulation was performed by CloudSim, and the results show that the performance of this algorithm is better than that of FCFS and RR algorithms. In [29], the authors used ACO for mobile cloud computing that requires specific resources. This method executed offloaded tasks in fog devices (FDs) by delay, complete-time, and energy consumption objectives. The time order of their simulation depends on the number of cycles, tasks, and ants.

### 2.3.3. PSO-Based Meta-Heuristic Algorithms

PSO was introduced by Kennedy and Eberhart in 1995 [30]. PSO is a population-based random optimization algorithm. In this algorithm, the dimension of the particles is equal to the number of tasks, and the position of the particle shows the mapping of VMs to tasks. Some of the scheduling schemes only use the basic PSO algorithm, but others use the improved model of this algorithm. Masdari et al. [31] presented a PSO algorithm based on workflow and project schedule for a cloud environment. Furthermore, classification is provided for the PSO algorithm based on the objective functions, features, and constraints.

Researchers in [14] present the MaOPSO to find a representative set of well-distributed non-dominated solutions close to the Pareto front for all objectives. This algorithm improves convergence and diversity compared to multi-objective methods.

Yassa et al. [32] proposed a scheme using hybrid PSO and HEFT. The algorithm aims to optimize makespan, cost, and power consumption. The algorithm starts by initializing the position and velocity of particles in PSO. The HEFT algorithm is applied several times to find an efficient solution to minimize the makespan. The results show that their approach is not only better in terms of cost and power consumption, but it also improves makespan.

The authors in [33] developed a hybrid approach for the recording of medical images. In their method, PSO is used to compute the Mutual Information (MI) using a weighted linear combination of image intensity and image gradient vector flow (GVF) intensity. Their proposed method was successfully tested in a lot of experiments and showed high accuracy and effectiveness.

### 2.3.4. Other Evolutionary Meta-Heuristic Algorithms

The authors of [5] propose a bio-inspired solution based on the Bees Life algorithm to solve the scheduling problem in the FC environment. This solution is based on the distribution of a set of tasks between all FDs. Analysis of the execution time of the CPU and allocated memory by FDs after simulation showed that this method performs better than PSO and GA.

In [34], the authors introduce a swarm-based meta-heuristics optimization method called Krill Herd (KH) and then accelerate its global convergence speed by a chaotic theory called CKH. The objective function in the KH is based on the minimum distance between the food position and the location of the krill [35]. To evaluate the results of the proposed method [34], various problems have been used. The results show that the performance of CKH with an appropriate chaotic map is better than that of other state-of-the-art meta-heuristic methods.

The researchers in [36] propose a quantum-inspired binary grey wolf optimizer (QI-BGWO) method for the unit commitment (UC) problem. The proposed method integrates the concepts of quantum computing with the BGWO to improve the hunting process of the wolf pack. The results show the effectiveness and improvement of the proposed method based on binary and quantum computations for the UC problem.

In [37], the cuckoo search and flower pollination algorithm as two meta-heuristic methods are used to solve different optimization problems. The results show that all platforms have similar performance on average. Additionally, if the algorithm is based on a balanced search and combines exploration with exploitation, it guarantees high-quality solutions.

In [38], the authors proposed an innovative algorithm for the group decision-making (GDM) problem with triangular neutrosophic additive reciprocal matrices. To evaluate GDM, they used triangular fuzzy numbers because of incompatibility with preferential relationships. By analyzing previous studies, they concluded that fuzzy preference relations have some drawbacks owing.

The improved whale optimization algorithm (IWOA) is presented in [39] for solving the 0–1 knapsack problems on a variety of scales. In this method, a penalty function is added to evaluate the performance of the solutions. The authors gained the trade-off between diversification and intensification through using two strategies: local strategy strategies (LSS)

and Lévy flight walks. The results show better performance for the proposed algorithm than other state-of-art algorithms.

In [40], an evolutionary multi-objective optimization algorithm is presented for high-performance computing in cyber–physical social systems (CPSS). To examine the feasibility of their proposed model, a floor planning case study is used. The B*-tree algorithm and the multi-step simulated annealing (MSA) algorithm are used to solve this problem. The performance of the proposed method was 74.44%. The researchers in [41] also present a multi-objective method based on a hybrid hitchcock bird algorithm and fuzzy signature (MOHFHB) for task scheduling in cloud computing. They improved makespan and resource utilization compared to both the Moth Search Algorithm with Enhanced Multi-Verse Optimizer and the Fuzzy Modified Particle Swarm Optimization. The authors of [42] also used Opposition-based learning to optimize PSO (OPSO) for task scheduling in a cloud computing environment. They improved the convergence of standard PSO, energy consumption, and makespan.

### 2.4. Hybrid Heuristic Algorithms

Guddeti et al. [43] presented a new bio-inspired algorithm (BIA) for job scheduling and resource management in the cloud computing environment. The proposed algorithm is a combination of modified PSO and CSO. The simulation was performed using PySim tools, and the results show that the proposed algorithm reduces the average response time and reduces resource utilization.

Delavar et al. [44] offered a mix of GA, best fit, and Round Robin (RR) algorithms to reduce the number of algorithm iterations. An optimal initial population is obtained by integrating the best Fit and RR algorithms. They optimized makespan, load balancing, and speed.

Kaur et al. [45] offered a mix of ACO and MAX–MIN algorithms (MMACO) to optimize load balancing and makespan. In the MAX–MIN algorithm, big tasks have higher priority than smaller tasks. For their proposed method, the waiting time is reduced for smaller tasks because small jobs run parallel on the fastest machines. The results of simulation in CloudSim show improved performance of the proposed algorithm.

Researchers in [46] provide a Hybrid Flamingo Search with a Genetic Algorithm (HFSGA) for cost-efficient QoS-aware task scheduling in a fog–cloud environment. This strategy improved the makespan and cost compared to ACO, PSO, GA, Min-CCV, Min-V, and RR algorithms.

A group of researchers in [47] worked on task scheduling in a cloud environment by an improved pathfinder algorithm using opposition-based learning (OBLPFA). Their approach improved total execution time, cost, and resource utilization compared to PSO, dragonfly, the Arithmetic Optimization Technique, the Reptile Search technique, the Aquila Optimization method, and Lion Optimization methods.

The hybrid strategy has also been used in [48] by combining a neural network-based method with heuristic policy (JNNHSP) to present an optimized task scheduling algorithm in a hierarchical edge cloud environment. The authors improved the scheduling error ratio, average service latency, and execution efficiency.

### 2.5. Hyper Heuristic Algorithms

Tsai et al. [8] presented a hyper-heuristic algorithm to find better scheduling solutions in the cloud computing environment. To evaluate the performance of the proposed algorithm, this method was implemented in the cloud and in the actual Hadoop system. The results show that the proposed algorithm significantly decreases makespan compared to other scheduling algorithms.

Gomez et al. [49] provide a multi-objective framework for hyper-heuristic selection to solve the two-dimensional bin-packing problem. The solution includes a multi-objective evolutionary learning process using genetic operators to generate a set of rules to represent the hyper-heuristic. This method minimizes the waste of space in the bin when allocating

pieces. Their case studies are a large set of bin-packing problems, including unordered convex and non-convex parts, based on different conditions and performance measures. The results indicate better solutions than those of single heuristics.

Chen et al. [50] proposed a hyper-heuristic framework and a high-level quantum-inspired learning strategy to improve the performance of the framework. Experimental results show that this method improved the search speed by 38%. The simulation results show that the proposed method has good performance compared to state-of-the-art methods such as HEFT, GA, and RH. Researchers in [51] present a two-stage technique based on the New Caledonian Crow Learning Algorithm and reinforcement learning strategy (PRLCC) for task scheduling in a cloud environment. Their evaluation in the CloudSim simulator showed some improvements in the waiting time, energy consumption, and resource utilization compared to other state-of-the-art methods.

The mentioned scheduling methods organized by scheduling type, technology design, and objectives parameters are categorized in Table 1.

**Table 1.** Categorization of task-scheduling strategies.

| Algorithm | Strategy | Scheduling Objectives | Environment | Pros and Cons |
|---|---|---|---|---|
| SABA [18] | Heuristic | Makespan, security, and budget | Cloud/Real environment | Improves response time. Ignores energy consumption. |
| MOHEFT [19] | Heuristic | Makespan and cost | Cloud/Real environment | Trade-off between cost and makespan. |
| EIPR [20] | Heuristic | Deadlines, total execution time, and budget | Cloud/Real environment | Improves performance. Ignores energy consumption. |
| Heuristic [21] | Heuristic | Makespan and execution cost | Cloud–Fog/CloudSim | Cost efficient. No scalability. |
| JLGA [23] | Meta-heuristic | Makespan and load balancing. | Cloud/MATLAB | Energy efficient. |
| RSS-IN [25] | Meta-heuristic | Latency and stability | Fog/MATLAB | Decreases latency. Ignores energy consumption. |
| ACO [28] | Meta-heuristic | Makespan | Cloud/CloudSim | Local optimum problem. |
| CMSACO [29] | Meta-heuristic | Delay, complete time, and energy consumption | Fog/Simulation | Ignores time complexity. |
| BLA [5] | Meta-heuristic | Execution time and memory size | Fog/C++ | Better performance than basic evolutionary algorithms. |
| MOHFHB [41] | Meta-heuristic | Makespan, resource utilization, energy consumption, latency, and degree load balance | Cloud/Simulation | Optimizes energy consumption and latency. |
| OPSO [42] | Meta-heuristic | Energy consumption and makespan | Cloud/CloudSim | Convergence of standard PSO, energy consumption, and makespan. |
| DVFS-MODPSO [32] | Hybrid-heuristic | Makespan, cost, and energy | Cloud/Real environment | Optimizes performance. |
| BIA [43] | Hybrid-heuristic | Response time and optimum usage of resources | Cloud/PySim | Resource efficient. Ignores energy consumption. |
| HSGA [44] | Hybrid-heuristic | Makespan and load balancing | Cloud/Real environment | Ignores time complexity. |
| MMACO [45] | Hybrid-heuristic | Makespan and load balancing | Cloud/CloudSim | Improves performance. |
| HFSGA [46] | Hybrid-heuristic | Makespan and cost | Fog–Cloud/MATLAB | Optimized for deadline-satisfied tasks. |
| OBLPFA [47] | Hybrid-heuristic | Execution time, cost, and resource utilization | Cloud/CloudSim | Improved time complexity. |
| JNNHSP [48] | Hybrid-heuristic | Service latency | Edge–Cloud/Real | Improves scheduling error ratio, average service latency, and execution efficiency. |
| HHSA [8] | Hyper-heuristic | Makespan and computation Time | Cloud/CloudSim and Hadoop | Realistic environment. Time efficient. |

## 3. The Proposed Approach

In this section, the system model and the scheduling method are presented.

### 3.1. System Model and Case Study

The nodes in the sensor networks receive data from their surroundings and send them to FDs through gateways. These data are either processed in FDs or sent to the cloud. A smart surveillance system is designed to coordinate multiple cameras for monitoring a specific area. Video surveillance/object tracking software is a collection of distributed mobile smart cameras.

The application includes six modules: the motion detector, object detector, object tracker, user interface, and pan, tilt, and zoom control (PTZ). The camera sends the video stream to the motion detector module. This module sends it to the object detector after applying a filter to the video stream. This module identifies the object's position and sends it to the tracker module; then, the appropriate PTZ is calculated and sent to the PTZ

control. Finally, a fraction of the video streams containing a traced object is sent to the user's device [52].

The FD and application properties are explained as follows.

### 3.1.1. FD

An FD is a micro data center that analyzes, filters, and stores data from sensors. The FD's properties include MIPS, RAM, up bandwidth, down bandwidth, the level number in the topology, rate per MIPS, power in the busy state, and idle power. Each FD includes hosts as $\{Host_1, Host_2, ..., Host_n\}$. The host properties include RAM, bandwidth, storage, and PEs. The total bandwidth of all hosts in each FD is between $FB_{Lower}$ and $FB_{Upper}$.

In an FD, PEs of hosts are allocated to application modules and are executed. The most important feature of a PE is MIPS. This value is set at the start of a simulation for all FDs. After allocation of a PE to an application module, the total allocated MIPS of all PEs is updated.

The total allocated MIPS of an FD can be expressed by $TAM = \sum_{i=1}^{N} \sum_{j=1}^{M} PEM_{ij}$ that is less than or equal to the MIPS of that FD ($TAM \leq FD_{MIPS}$). $N$ is the number of hosts in the FD, $M$ is the number of PEs in a host, and $PEM_{ij}$ is the MIPS of the *jth* PE in the *ith* host.

### 3.1.2. Application

An application includes some modules. These modules are related by edges. Data as tuples are transferred between two modules by tuple mapping.

- **Application module:** This module is a type of VM. The module's properties include MIPS, size, bandwidth, and the number of PEs. The number of modules in each FD is more than the number of PEs $\sum_{i=1}^{C} Module_i > \sum_{j=1}^{K} FD_j$, where $C$ is the total number of modules, and $K$ is the total number of FDs. The application modules of the considered case study include an object detector, motion detector, object tracker, and user interface.

- **Application edge:** The application modules are connected by edges. Each application edge is between two modules. In fact, tuples are transferred between modules by edges. Each edge has two important features: CPU length and data size. In fact, $\sum_{i=1}^{M} TCL_i \leq MIPS_{module}$, and $\sum_{i=1}^{M} DS_i \leq RAM_{module}$. This means the total CPU length and the data size of all input tuples to a module must be less than or equal to the MIPS and RAM capacity of that module. $TCL_i$ is the total CPU length, and $DS_i$ is the data size of the *ith* tuple. $M$ is the total number of tuples. $MIPS_{module}$ is the module's MIPS.

- **Application tuple mapping:** The tuple is the input/output relationships of the application modules that send data from one module to another module ($module_i$ to $module_j$; $i \neq j$).

- **Application loop:** Each workflow of modules is an application loop. Each application has some workflow that connects modules by edges.

### 3.2. HHS

As in Figure 1, a case study is an application with several modules as $\{M_1, M_2, ..., M_c\}$. When the application starts, then a number of modules must be executed in the FDs as $\{Fog_{11}, Fog_{12}, ..., Fog_{Root}\}$ by PEs as $\{PE_{11}, PE_{12}, ..., PE_{nb}\}$. The proposed algorithm in the HHS box is used to allocate the best PEs to modules.

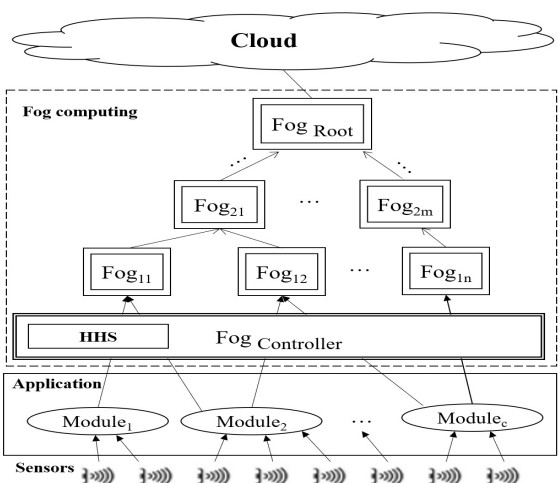

**Figure 1.** Allocation of PEs to modules in FDs.

First of all, the low-level heuristics $\{H_1, H_2, H_3, H_4\}$ in a repository are placed so that $H_i$ is GA, PSO, ACO, and SA, respectively. The proposed algorithm uses data mining as in Section 3.2.7. The best heuristic is selected among the candidate algorithms for the new workflow. The HHS algorithm steps are shown in Figure 2.

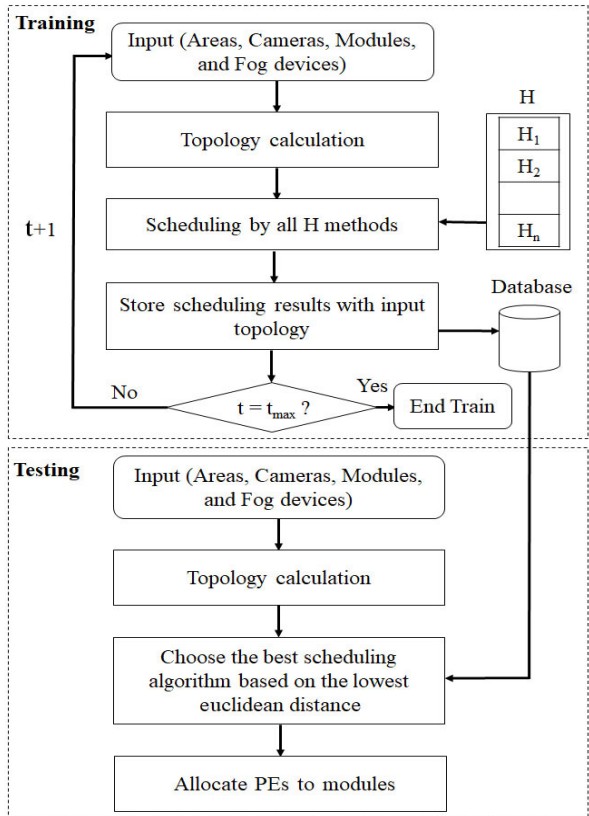

**Figure 2.** HHS algorithm flowchart.

### 3.2.1. Encoding Individual

In the proposed method, each chromosome is a scheduling solution. A chromosome includes N genes. Each gene represents a PE. The chromosome length is equal to the number of PEs. The value of each gene shows the module ID. In this sample chromosome, $PE_1$ allocates to $module_4$, $PE_2$ allocates to $module_1$, $PE_3$ allocates to $module_M$, and $PE_N$

allocates to $module_3$. In the low-level heuristics in this paper, the population is randomly initialized by several chromosomes.

### 3.2.2. Fitness Function

In the following, first, the low-level heuristics of the HHS algorithm are presented, which include GA, PSO, ACO, and SA, and then the proposed algorithm is explained. These algorithms use the fitness function to evaluate the proper allocation of PEs to modules. The fitness function [11] can be defined by $Fitness = \frac{1}{w_1 * \sum_{i=1}^{N} U_i + w_2 * BW}$, where $BW$ denotes the bandwidth in a module. $w_1$ and $w_2$ are weights of $TUC$ and $BW$, respectively, and their values are equal 0.5, $U_i$ is the utilization of $ith$ tuple, and $N$ is the total number of tuples. The value of $BW$ for each module is set in the start of the simulation. To find the best scheduling, the fitness value must be minimized.

### 3.2.3. Total Execution Cost

Total execution cost [11] can be calculated by $PEC + (CC - LUUT) * RPM * LU * TM$, where MIPS (Million Instruction Per Second) of allocated PEs are obtained in the calculated time frame. The time frame is different from the simulation's current time and last utilization time. $PEC$ is the past execution cost, $CC$ is the CloudSim clock or current time of simulation, $LUUT$ is the last utilization update time, $RPM$ is the rate per MIPS, which is different for each inter-module edge, and $TM$ is the total MIPS of the host. $LU$ is the last utilization ($LU$), which is calculated as $LU = Min(1, TMA/TM)$, and $TMA$ is the total allocated MIPS of the host.

### 3.2.4. Total Network Usage

Total network usage [11] can be expressed by $\frac{\sum_{i=1}^{N}(TL_i * TS_i)}{MST}$. The input/output relationships between modules are defined by tuples. The usage of network resources depends on the size of transferred tuples at a certain time. $TL_i$ and $TS_i$ are the total latency and the total size of $ith$ tuple, $N$ is the total number of tuples, and $MST$ is the maximum simulation time.

### 3.2.5. Energy Consumption

The energy consumption [11] can be given by $CEC + (NT - LUUT) * HP$. The FD's energy consumption is calculated by the power of all hosts in a certain time frame of execution. $CEC$ is the current energy consumption, $NT$ is the now time, $LUUT$ is the last utilization update time, and $HP$ is the host power in $LU$.

### 3.2.6. Application Loop Delay

The application loop delay can be expressed by the CloudSim clock and the tuple's end time. $T1$ is the tuple start time, $T2$ is the tuple type to average CPU time, $CC$ is the CloudSim clock, $(CC - T1)$ is the execution time, and $C1$ is the number of executed tuple types. The tuple's end time [11] can be expressed by $CC - T1$ when $T2$ is calculated, or $\frac{T1*C1+(CC-T1)}{C1+1}$ when $T2$ is not calculated. The application loop delay is calculated by $CC - ET$. $CC$ is the CloudSim clock, and $ET$ is the emitting time of a tuple. $ET$ is calculated by sending the time of a module to another module. The tuple receipt time is calculated by $\frac{T1*C2+(CC-ET)}{C1+1}$. $C2$ is the number of received tuple types.

The proposed method consists of two phases: training and testing. The pseudocode of the proposed approach is Algorithm 1.

---

**Algorithm 1** HHS.

---

　　**Input: number of areas, number of cameras, scheduling methods.**

1: Initialization of number of areas, number of cameras, scheduling method.
2: **for** $area_i = 1$ to $A$ **do**
3: 　　**for** $camera_i = 1$ to $C$ **do**
4: 　　　　Task scheduling by GA, PSO, ACO, and SA, respectively.
5: 　　　　Save the results in a dataset file.
6: 　　**end for**
7: **end for**
8: Read new workflow.
9: **for** $i = 1$ to $M$ **do**
10: 　　$D_i$ = Euclidean distance of new sample with dataset row $i$.
11: **end for**
12: The best scheduling algorithm = MinDistance($D$).
13: Execute the application.
14: Calculate energy consumption, network usage, execution time, and total cost using Sections 3.2.3, 3.2.4, 3.2.5, and 3.2.6.

---

- **Training phase:** Initially, 64 different workflows enter the system. The proposed algorithm includes GA [22], PSO [30], ACO [26], and SA [53] and is implemented to allocate PEs to modules in all workflows and for the intelligent monitoring system that comes along with the modules. The energy consumption, network usage, and total execution cost of each algorithm are achieved for each workflow. Then, the results are stored in the database, and for each workflow, the best algorithm is selected.
- **Testing phase:** A new workflow enters the system. Then, the Euclidean distance between the new workflow and examples inside the database is obtained. The best algorithm is chosen. Then, the energy consumption, network usage, and total execution cost of the new workflow are calculated. Finally, the results are returned.

3.2.7. Data Mining

To find the best low-level heuristic for scheduling, a data mining method is implemented. In this method, first, sample topologies are entered into the fog network as training data; then, scheduling methods are performed on them. Second, a new topology is entered; then, that is scheduled by the best low-level heuristic based on the training phase results. These training and testing phase are as follows.

**Training phase**: In this part of HHS, as described in Section 3.2, a database is created with some features. Each of the rows of these data has six columns, which include the number of areas, the number of cameras, the energy consumption, the total network usage, the total execution cost, and the type of scheduling algorithm. In fact, these columns include the network topology and output parameters generated by the scheduling algorithms. Since the number of areas and cameras is considered to be between 1 and 4, there are 16 different modes. Additionally, by 4 algorithms, GA, PSO, ACO, and SA create 64 samples and save them to the database.

**Testing phase**: In this part, according to the network topology (the number of areas and cameras), a quick search is made, and a row of training samples that has the least Euclidean distance as $\sqrt{(A2 - A1)^2 + (C2 - C1)^2}$ with the input topology is selected; the last column is run as the scheduling algorithm. A1 and A2 are the number of areas in the training and testing phase, respectively; also, C1 and C2 are the number of cameras in the training and testing phase, respectively, of the proposed method. The result of this equation is used to compare the distance between the input topology in the testing phase and the topology samples in the training database.

3.2.8. Algorithm Parameters and Complexity Analysis

The main parameters of the low-level heuristics and HHS are A as the number of areas, C as the number of cameras, EC is energy consumption, TNU is total network usage, TEC is total execution cost, and S is the scheduling algorithm as $H_i$ or HHS. These values are initialized at the start of their algorithms. Additionally, their computational complexities are presented in Table 2 and are explained as follows.

- In GA, the fitness calculates in $O(nm)$ so that $n$ is the number of individuals with size $m$. The crossover and mutation operators calculate in $O(nm)$. The elitism order is $O(nm)$. The computational complexity of GA is $O(gmn)$.
- In PSO, the algorithm gets the position and velocity of all particles calculated in $O(n)$. The fitness value for each particle calculates in $O(m)$, and $m$ is the particle size. The computational complexity of PSO is $O(gmn)$.
- In ACO, the pheromones update in $O(k)$. Since the upper bound of $O(k)$ is $O(n)$, the computational complexity is $O(gmn)$. The computational complexity of ACO is $O(gmn)$.
- In SA, the fitness of each particle and a new particle calculate in $2 * O(m)$. The computational complexity of GA is $O(gm)$.
- In HHS, $k$ is the size of topology samples in the database. Additionally, the computational complexity of HHS depends on the algorithm selected based on Euclidean distance.

In the mentioned algorithms, $g$ is the number of iterations in all the above orders. All iterations execute in $O(g)$; $n$ is the size of the population (or particles or ants). All population members process in $O(n)$; $m$ is the individual size. An individual processes in $O(m)$.

**Table 2.** Algorithm parameters and complexity analysis.

| Algorithm | Parameters | Complexity |
|---|---|---|
| GA | Mutation rate = 0.5<br>Crossover rate = 0.9<br>Elitism = 10% | $O(g * (n * m + n * m + n)) = O(gnm)$ |
| PSO | Swarm size = 10<br>Acceleration rate = 2 | $O(g * n * (m + m)) = O(gnm)$ |
| ACO | Ant count = 10<br>Pheromone updating rate = 0.1<br>Choosing probability = 0.85<br>Influence weights = 0.95 | $O(g * n * (m + k)) = O(gnm)$ |
| SA | Mutation rate = 0.3<br>Starting temperature = 1<br>Cooling rate = 0.05 | $O(g * (m + m)) = O(gm)$ |
| HHS | Training samples = 64<br>Testing samples = 16 | $O(k + O(SelectedAlgorithm)) = O(SelectedAlgorithm)$ |

## 4. Evaluation

In this section, the performance of the proposed scheduling method as HHS is analyzed and compared with traditional and heuristic methods (FCFS, concurrent, and DP) [12] and meta-heuristic methods (NSGA-III [13] and MaOPSO [14]).

### 4.1. Experimental Environment

The experimental environment includes Intel Core i5 CPU, 3 GB memory, a 500 GB HD, Windows 10 32-bit operating system, Netbeans, JDK8.0, and iFogSim [52]. The program iFogsim is a Java-based library for simulating an FC environment and is used to simulate

the workflow scheduling problem. The implementation inherits and extends some of the iFogsim classes, such as FogDevice, Controller, and DCNS. The CreateFogDevice and CreateApplication functions of the DCNS class are updated to create FDs and application properties. Additionally, the UpdateAllocatedMIPS function of the FogDevice class for allocation of PEs to modules is updated.

The initial population size is 64 for the training phase and 16 for the test phase. The number of checked areas changes from 1 to 4. Other parameters of the scheduling algorithms (Table 2) are as follows. In GA, mutation rate = 0.5, crossover rate = 0.9, and elitism = 10%. In PSO, swarm size = 10, and acceleration rate = 2. In ACO, ant count = 10, pheromone updating rate = 0.1, choosing probability = 0.85, and influence weights = 0.95. In SA, mutation rate = 0.3, starting temperature = 1, and cooling rate = 0.05. In HHS, the number of training samples = 64, and the number of test samples = 16. Each area has 1 to 4 smart cameras that monitor the area. These cameras connect to an area gateway that is responsible for accessing the Internet. Based on the above configuration, the physical topology is designed. In this topology, the cloud is at the highest level, and areas, cameras, and other FDs are at the network edge.

### 4.2. Simulation Configuration

As shown in Table 3, each FD, as a micro data center, has many parameters, including MIPS, RAM (kilobyte), UpBW (up bandwidth by kilobyte per second), DownBW (down bandwidth by kilobyte per second), level in the hierarchical topology, rate per MIPS, and busy and idle power (watts). The application module only has a bandwidth feature, which is set in the UpBW column.

**Table 3.** Simulation configuration.

| Name | MIPS | RAM | UpBw | DownBw | Level | RatePerMips | Busy Power | Idle Power |
|---|---|---|---|---|---|---|---|---|
| FD | 44,800 | 40,000 | 100 | 10,000 | 0 | 0.01 | $16 * 103$ | $16 * 83.25$ |
| Area's FD | 2800 | 4000 | 10,000 | 10,000 | 1 | 0 | 107.339 | 83.4333 |
| Camera's FD | 500 | 1000 | 10,000 | 10,000 | 3 | 0 | 87.53 | 82.44 |
| Application module | 1000 | 10 | 1000 | - | - | - | - | - |

Further, the case study configurations of inter-module edges are as follows (CPU length, tuple length): raw video stream (1000, 2000), motion video stream (2000, 2000), detected object (500, 2000), object location (1000, 100), and PTZ parameters (100, 100). Each sensor sends 20,000 bytes with 1000 MIPS to the application with an average interval time of 5 milliseconds.

### 4.3. Statistical Analysis of Fog-Based Case Study

The number of executed modules has been obtained according to different experiments. Figure 3 shows total executed modules versus FD numbers. The horizontal axis shows the number of areas, cameras, and FDs. The allocation of a large number of modules to FDs need too many processes by PEs; because of this, an optimization algorithm is used to quickly search the space for answers.

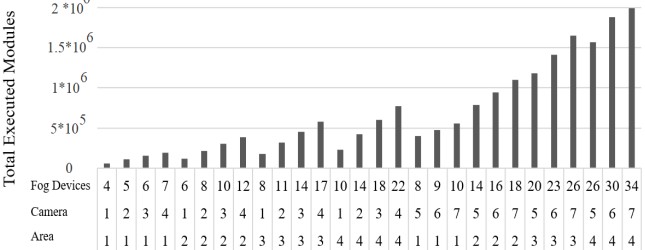

**Figure 3.** Total executed modules versus FD numbers.

### 4.4. Analysis Based on the Number of Users

The energy consumption, total execution cost, and delay of the application loop for 1 to 10 users is shown. Table 4 shows that the average energy consumption of the HHS algorithm is better than those of FCFS by 7.1%, concurrent by 28.6%, and DP by 2.7%. Based on Table 5, the average total execution cost of the HHS algorithm is better than those of FCFS by 53.48%, concurrent by 65.26%, and DP by 52.14%. In Table 6, the delay of HHS algorithm is better than those of FCFS by 6%, concurrent by 28%, and DP by 52.14%.

**Table 4.** Energy consumption statistics of FCFS, Concurrent, DP, and HHS based on the number of users. Avg = $Average * 10^{-7}$, Max = $Maximum * 10^{-7}$, Min = $Minimum * 10^{-7}$, and SD = $StandardDeviation * 10^{-3}$.

| Value | FCFS | Concurrent | DP | HHS |
|-------|------|------------|------|------|
| Avg | 1.54 | 2.00 | 1.49 | 1.43 |
| Max | 1.54 | 2.15 | 1.52 | 1.43 |
| Min | 1.54 | 1.69 | 1.46 | 1.43 |
| SD | 1.97 | 1630 | 216 | 4.90 |

**Table 5.** Total execution cost statistics of FCFS, Concurrent, DP, and HHS based on the number of users. Avg = $Average * 10^{-6}$, Max = $Maximum * 10^{-6}$, Min = $Minimum * 10^{-6}$, and SD = $StandardDeviation * 10^{-3}$.

| Value | FCFS | Concurrent | DP | HHS |
|-------|------|------------|------|------|
| Avg | 2.89 | 3.87 | 2.81 | 1.34 |
| Max | 2.90 | 4.53 | 2.87 | 1.35 |
| Min | 2.89 | 3.18 | 2.74 | 1.33 |
| SD | 2.79 | 463 | 41.7 | 6.95 |

**Table 6.** Application loop delay statistics for FCFS, Concurrent, DP, and HHS based on the number of users.

| Value | FCFS | Concurrent | DP | HHS |
|-------|------|------------|------|------|
| Avg | 107 | 139 | 103 | 100 |
| Max | 107 | 155 | 106 | 103 |
| Min | 107 | 117 | 101 | 96 |
| SD | 16.8 | 11.4 | 1.52 | 2.61 |

In another comparison, the number of users in a fog-computing architecture is considered. The traditional and heuristic methods are presented as [13]. Figure 4a is based on energy consumption, so the HHS algorithm has the minimum energy consumption by $1.43 * 10^7$. Figure 4b shows the total execution cost; that of the HHS algorithm is $1.34 * 10^6$. Thus, it is better than the FCFS, concurrent, and DP algorithms. According to Figure 4c, the minimum delay of the HHS algorithm is 96 s; thus, the delay of the application loop is optimized by HHS.

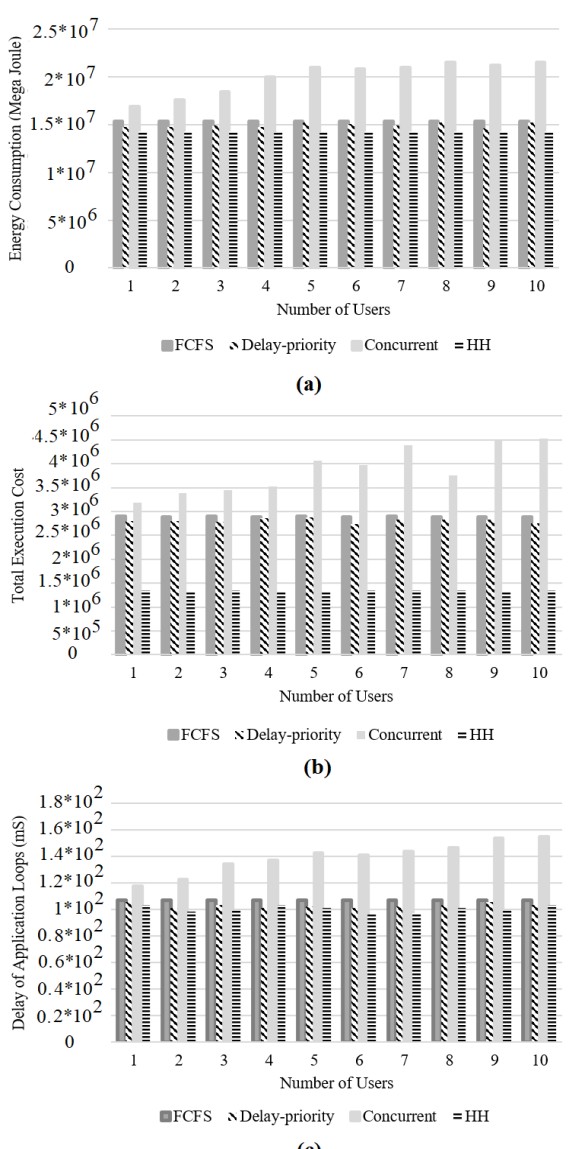

**Figure 4.** Comparison of scheduling algorithms based on number of users: (**a**) Energy consumption, (**b**) Total execution cost, and (**c**) Delay of application loop.

*4.5. Analysis Based on the Number of Devices*

This section shows the energy consumption and total network usage based on different numbers of fog devices. Additionally, the proposed approach is compared with the other meta-heuristic methods.

### 4.5.1. Energy Consumption

As Figure 5 shows, HHS after GA has the least average energy consumption; also, the minimum energy consumption of GA and HHS is $1.36 * 10^7$. The highest energy consumption is $1.54 * 10^7$ megajoules, which is related to the ACO algorithm with 4 areas and 4 cameras with 22 FDs.

The lowest energy consumption (megajoules) is $1.36 * 10^7$ for the GA and HHS algorithms with one area, one camera, and four FDs. The average energy consumption by various algorithms is as follows: the GA algorithm, $1.38 * 10^7$; the PSO, $1.51 * 10^7$; ACO, $1.52 * 10^7$; SA, $1.51 * 10^7$; and HHS, $1.40 * 10^7$.

The proposed algorithm decreases the average energy consumption of PSO by 7.17%, ACO by 7.42%, and SA by 6.91%. Hence, the energy consumption of the proposed algorithm in different modes is less than that of SA, ACO, and PSO; thus, HHS has better performance

than the three mentioned algorithms. The energy consumption by GA and HHS are almost the same, but GA has a slightly smaller value.

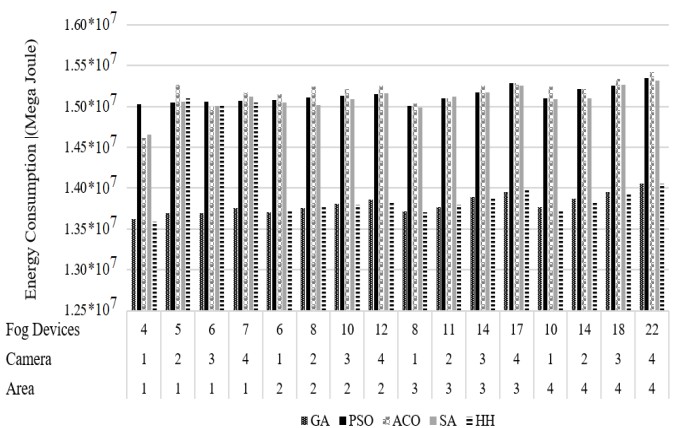

**Figure 5.** Energy consumption of GA, PSO, ACO, SA, and HHS.

### 4.5.2. Total Network Usage

The graph of total network usage is shown in Figure 6. With increasing areas and the number of FDs, the network usage increases. Moreover, the proposed scheduling algorithms have fully utilized all the given network resources. The network usage of resources is the same for all algorithms at the end of the simulation. Additionally, the network usage statistics for all algorithms are approximately equal; thus, the average is $2.6 * 10^5$, the maximum is $6.65 * 10^5$, the minimum is $4.17 * 10^4$, and the standard deviation is $1.72 * 10^5$.

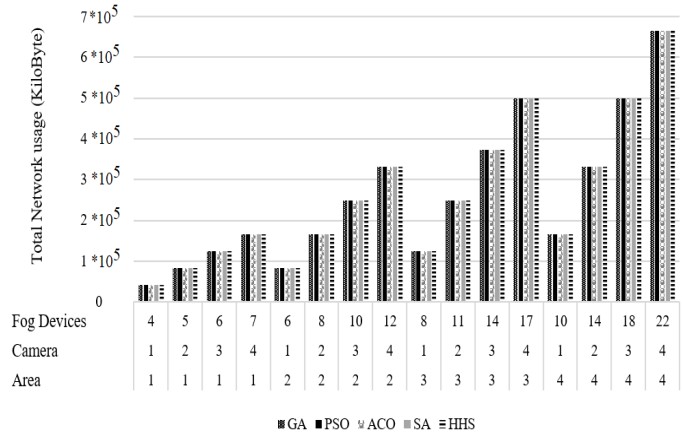

**Figure 6.** Total network usage of GA, PSO, ACO, SA, and HHS.

### 4.5.3. Comparison with Meta-Heuristic Methods

The HHS algorithm is compared with two meta-heuristic methods: NSGA-III [13] and MaOPSO [14]. The chromosome encoding is as described in Section 3.2.1. The parameters of these two algorithms are as follows.

- NSGA-III: population size = 100, crossover probability = 0.9, mutation probability = 0.5, and max iterations = 50.
- MaOPSO: swarm size = 100, archive size = 100, mutation probability = 0.5, and max iterations = 50.

The average energy consumption and total execution cost of MaOPSO and NSGA-III are better than that of HHS. Of course, MaOPSO and NSGA-III reduce the energy consumption and total execution cost with a great deal of time, which is not a good result for an IoT application. The delay of the application loop is decreased by HHS compared to

MaOPSO by 28.1% and compared to NSGA-III by 40.8%. Therefore, reducing the delay is an advantage relative to the MaOPSO and NSGA-III algorithms.

The results in Figure 7a,b show that the energy consumption and the total execution cost of NSGA-III and MaOPSO are less than that of the HHS algorithm. This is due to the use of low-level algorithms in HHS. NSGA-III and MaOPSO are good at finding the optimal answer, but as Figure 7c shows, their application loop delay is greater than that of HHS. Since many IoT applications need to execute in real-time, thus HHS algorithm with its low delay is more suitable.

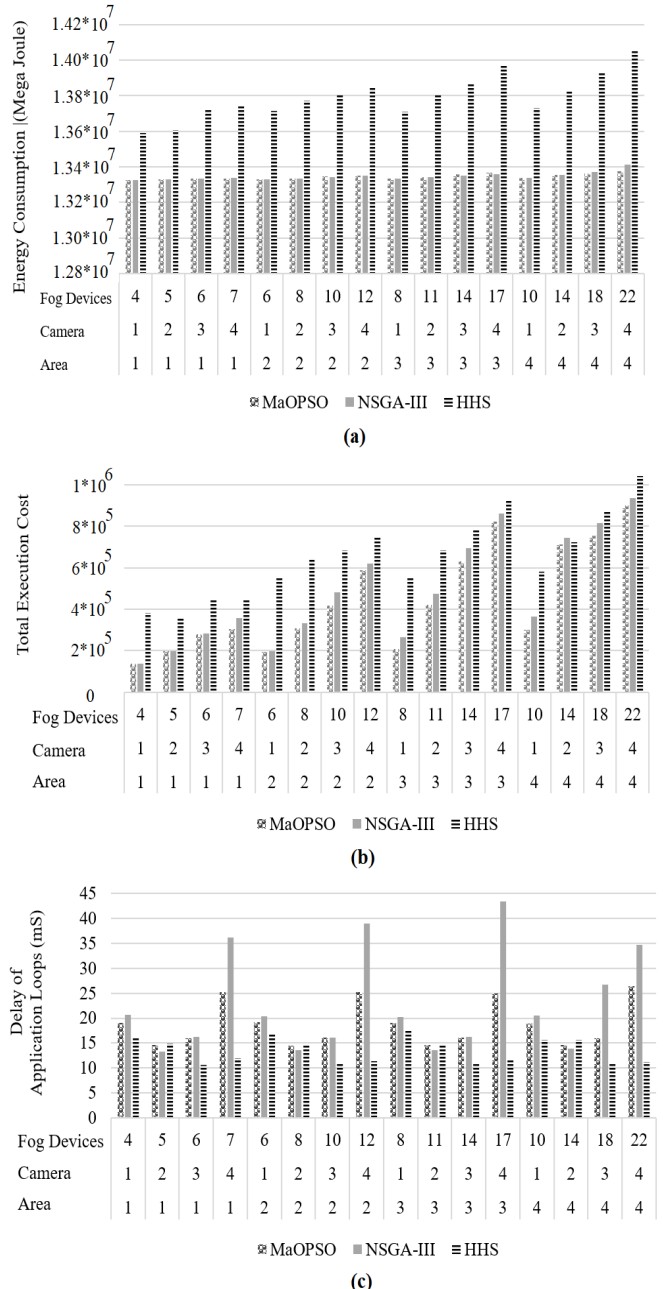

**Figure 7.** Comparison of HHS algorithm to MaOPSO and NSGA-III: (**a**) Energy consumption (**b**) Total execution cost, and (**c**) Delay of application loop.

#### 4.5.4. Execution Time

The simulation execution time for three different configurations is measured and can be seen in Table 7. After executing different algorithms on these configurations, the execution time of the simulation is obtained for each algorithm.

**Table 7.** Simulation execution times (seconds) for scheduling algorithms. (Con.: concurrent; MO: MaOPSO; Avg.: average).

| A | C | GA | PSO | ACO | SA | MO | NSGA-III | Con. | FCFS | DP | HHS |
|---|---|-----|-----|------|-----|------|------|------|------|------|------|
| 1 | 1 | 10.04 | 2.09 | 10.01 | 1.15 | 16.04 | 16.48 | 2.57 | 2.38 | 1.28 | 1.36 |
| 1 | 2 | 19.05 | 6.17 | 42.47 | 4.08 | 31.02 | 34.01 | 7.19 | 5.39 | 4.12 | 4.39 |
| 1 | 3 | 75.98 | 23.42 | 157.03 | 14.55 | 80.27 | 84.37 | 24.90 | 20.74 | 18.71 | 14.86 |
| 1 | 4 | 115.20 | 32.19 | 180.72 | 25.61 | 90.18 | 105.39 | 53.38 | 47.01 | 40.75 | 25.93 |
| 2 | 1 | 17.20 | 5.13 | 36.28 | 3.53 | 25.83 | 28.10 | 5.72 | 4.01 | 3.81 | 3.84 |
| 2 | 2 | 66.13 | 26.03 | 130.39 | 10.11 | 70.28 | 75.43 | 26.53 | 25.01 | 20.19 | 15.49 |
| 2 | 3 | 81.02 | 30.69 | 160.03 | 30.66 | 84.93 | 87.20 | 50.16 | 41.02 | 38.02 | 30.97 |
| 2 | 4 | 90.02 | 43.75 | 198.26 | 45.03 | 91.05 | 98.35 | 62.93 | 54.07 | 48.30 | 41.61 |
| 3 | 1 | 70.29 | 21.14 | 120.39 | 12.94 | 74.09 | 78.10 | 22.73 | 20.15 | 18.13 | 12.05 |
| 3 | 2 | 85.39 | 34.28 | 178.20 | 34.01 | 90.12 | 94.42 | 55.01 | 44.20 | 39.14 | 34.32 |
| 3 | 3 | 82.59 | 37.05 | 181.33 | 41.50 | 85.53 | 90.10 | 55.42 | 47.01 | 36.18 | 41.81 |
| 3 | 4 | 90.44 | 43.01 | 192.85 | 52.39 | 93.30 | 101.24 | 74.02 | 65.40 | 60.13 | 52.88 |
| 4 | 1 | 120.29 | 35.70 | 184.22 | 28.10 | 93.38 | 97.01 | 44.02 | 36.51 | 30.11 | 38.31 |
| 4 | 2 | 81.20 | 40.77 | 170.41 | 40.01 | 84.30 | 89.02 | 56.39 | 49.31 | 42.02 | 40.31 |
| 4 | 3 | 92.10 | 44.58 | 195.01 | 55.09 | 95.33 | 97.05 | 72.29 | 66.20 | 59.02 | 55.40 |
| 4 | 4 | 96.22 | 47.06 | 200.74 | 72.03 | 91.44 | 98.05 | 92.39 | 88.04 | 79.06 | 72.34 |
| 2 | 6 | 103.92 | 53.02 | 227.30 | 86.22 | 108.34 | 115.73 | 105.40 | 93.84 | 80.11 | 86.33 |
| 2 | 7 | 130.88 | 69.32 | 244.07 | 93.21 | 142.01 | 155.20 | 110.36 | 101.55 | 98.22 | 93.33 |
| 3 | 6 | 122.04 | 59.20 | 231.40 | 90.11 | 114.19 | 125.03 | 113.92 | 105.35 | 95.04 | 90.34 |
| 3 | 7 | 140.59 | 64.27 | 255.38 | 112.19 | 140.20 | 149.01 | 150.44 | 134.06 | 127.47 | 112.40 |
| 4 | 6 | 163.41 | 75.99 | 280.31 | 130.27 | 173.93 | 180.55 | 161.30 | 140.25 | 139.90 | 130.58 |
| 4 | 7 | 171.48 | 153.59 | 306.10 | 150.31 | 217.20 | 222.09 | 180.77 | 166.02 | 160.05 | 150.62 |
| Avg. | | 92.07 | 43.11 | 176.50 | 51.50 | 95.13 | 101.00 | 69.45 | 61.71 | 56.35 | 52.25 |
| Max | | 171.48 | 153.59 | 306.1 | 150.31 | 217.20 | 222.09 | 180.77 | 166.02 | 160.05 | 150.62 |
| Min | | 10.04 | 2.09 | 10.01 | 1.15 | 16.04 | 16.48 | 2.57 | 2.38 | 1.28 | 1.36 |
| SD | | 41.06 | 30.82 | 72.94 | 41.78 | 44.07 | 45.72 | 49.18 | 44.89 | 43.74 | 41.43 |

The average execution time of PSO of 43.11 s is less than that of HHS, but the energy consumption, network usage, and total execution cost of PSO are more than that of HHS. Additionally, ACO has the maximum value of the average execution time. As result, the HHS algorithm is better than other methods in all metrics and averages.

After all analyses, it is specified that the proposed approach is commensurate for an IoT application. Based on the time and resource sensitivity of this kind of application, low-level heuristic methods are effective. The evaluation results show the algorithm is able to get rid of some common issues in evolutionary algorithms; e.g., local optima can be solved by selecting the low-level methods. In fact, HHS selected the best algorithm to present an optimized scheduling strategy. Different metrics were tested to prove this claim, such as energy consumption, total execution cost, network usage, delay, and the number of users and devices. In comparison to basic scheduling algorithms, the HHS has better performance in energy consumption, total execution cost, network usage, and delay.

#### 5. Conclusions and Future Work

This paper presents an HHS algorithm by low-level heuristics based on data mining for task scheduling in an FC architecture. After a study of some meta-heuristic algorithms, the proposed method shows better results than other methods. The HHS algorithm reduces simulation time and increases decision-making power; assigning resources with specific

constraints to users is increased according to the type of workflow. The delay of the application loop is decreased in HHS compared to MaOPSO by 28.1%, NSGA-III by 40.8%, FCFS by 6%, concurrent by 28%, and DP by 52.14%. The average energy consumption is decreased in HHS compared to FCFS by 7.72%, concurrent by 11.69%, and DP by 3.63%. The average total execution cost is decreased in HHS compared to FCFS by 53.47%, concurrent by 55.90%, and DP by 45.28%. The average energy consumption and total execution cost of HHS are not better than those of MaOPSO or NSGA-III. Of course, MaOPSO and NSGA-III take a great deal of time to reduce energy consumption and total execution cost, which is not a good result for an IoT application. Therefore, reducing the delay is an important advantage over the MaOPSO and NSGA-III algorithms.

For future work, investigation of classification methods for finding appropriate candidate heuristics by adjusting the parameters of these algorithms in accordance with data and resource streams can be useful. Additionally, research on more IoT case studies with various topologies of FCs for medical care, smart homes, and vehicle transportation systems can be valuable. Latency and low power as two major constraints in the IoT require further research. Additionally, scheduling of application modules according to fault-tolerance, QoS requirements, and security overhead is an important issue in FC research.

**Funding:** This research received no external funding.

**Data Availability Statement:** Not applicable.

**Conflicts of Interest:** The author declares no conflict of interest.

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
