# Peer review of "Analyzing Meta-Heuristic Algorithms for Task Scheduling in a Fog-Based IoT Application"

_algorithms, doi:10.3390/a15110397_

Round 1
Reviewer 1 Report
The subject is topical. The paper is well written. The idea of ​​selecting a low-level heuristic method through data mining techniques is interesting and is shown to bring increased performance. The experiments are consistent and edifying.
More details on the scheduling scheme would have been desirable (section 3.1.1.). The diagram in figure 1 is unclear. Only 2 hosts appear and from Fog2* you go directly to Fogtk. More attention to this figure would not hurt. Then, figure 1 does not represent the scheduling scheme but the context of the application of the proposed Hyper-Heuristic Scheduling method.
Complexity analysis from section 3.5.1. must be reviewed. For example, related to PSO it is stated "In PSO, getting the position and velocity of all particles calculated in O(n). The fitness value for each particle calculated in O(m). m is the particle size." What is the complexity of the algorithm? At ACO (below) the complexity is given, but it is not clear where it results from, and so on.
Author Response
Reviewer#1, Concern # 1: The subject is topical. The paper is well written. The idea of ​​selecting a low-level heuristic method through data mining techniques is interesting and is shown to bring increased performance. The experiments are consistent and edifying.
Author response: Thank you for your comment.
Author action:
Reviewer#1, Concern # 2: More details on the scheduling scheme would have been desirable (section 3.1.1.). The diagram in figure 1 is unclear. Only 2 hosts appear and from Fog2* you go directly to Fogtk. More attention to this figure would not hurt. Then, figure 1 does not represent the scheduling scheme but the context of the application of the proposed Hyper-Heuristic Scheduling method.
Author response: Thank you very much for this comment.
Author action: This figure is modified. The explanations are reorganized and updated.
Reviewer#1, Concern # 3: Complexity analysis from section 3.5.1. must be reviewed. For example, related to PSO it is stated "In PSO, getting the position and velocity of all particles calculated in O(n). The fitness value for each particle calculated in O(m). m is the particle size." What is the complexity of the algorithm? At ACO (below) the complexity is given, but it is not clear where it results from, and so on.
Author response: I appreciate this comment.
Author action: The complexity analysis section is updated. Also, more details are shown in Table 2.

Reviewer 2 Report
It is not totally clear about the research focus. Is it only task allocation, or only task sequencing or both? Normally speaking, after allocating tasks to resources, it is quite often to consider the sequencing as well. It is vital to make it clear.
However, the HHS algorithm is unclear, and it would be helpful to incorporate a better explanation, preferably using a simple example.
Moreover, the research contribution in this paper must be made clear as no new algorithm is developed but the improved or modified algorithm is proposed, i.e. incremental contribution but not fundamental contribution.
What is the difference between scheduling in cloud computing and fog?
The literature reviews are not properly done in this paper. It’s also important if the author can summarize the advantages, and disadvantages of each strategy in a table. Authors have missed some state of the art schedule mechanisms that would add to the novelty aspect of the paper. To that authors are suggested to compare papers such as
An improved pathfinder algorithm using opposition-based learning for tasks scheduling in cloud environment
Hybrid heuristic algorithm for cost-efficient QoS aware task scheduling in fog–cloud environment
Multi-objective scheduling technique based on hybrid hitchcock bird algorithm and fuzzy signature in cloud computing
Combining neural network-based method with heuristic policy for optimal task scheduling in hierarchical edge cloud
A two-stage scheduler based on New Caledonian Crow Learning Algorithm and reinforcement learning strategy for cloud environment
Opposition-based learning inspired particle swarm optimization (OPSO) scheme for task scheduling problem in cloud computing
The reference of each equation should be written right above it.
In Section 3.3, fitness function is very simple. Why?
Section 3.5 should be explained in detail.
In Section 4, the performance evaluation of the proposed method is too simple, and it lacks the comparison with other advanced algorithms proposed in recent three years.
How set Table1?
The experiment is not comprehensive and thorough. For example, what is the overhead of the proposed algorithms? Are the algorithms still effective as the scale of data increases?
It is important to make it clear how other benchmarking methods are selected/employed when making comparisons against the proposed algorithm. More details are expected, otherwise, the comparison results may not be convincing. If benchmarking methods are not fine-tuned against the current problem, it means that these methods cannot perform at their very best, hence, the comparison is biased, i.e., in favor of the proposed method.
The reason for the strong and weak performance of the methods should be explained (more discussion) in the results section (Section 4).
Author Response
Reviewer#2, Concern # 1: It is not totally clear about the research focus. Is it only task allocation, or only task sequencing, or both? Normally speaking, after allocating tasks to resources, it is quite often to consider the sequencing as well. It is vital to make it clear.
Author response: Thank you for this comment.
Author action: The definition of scheduling has been highlighted in the introduction.
Reviewer#2, Concern # 2: However, the HHS algorithm is unclear, and it would be helpful to incorporate a better explanation, preferably using a simple example.
Author response: I appreciate this comment.
Author action: The subsections are reorganized for better explanations. An algorithm has been also added for more clarification.
Reviewer#2, Concern # 3: Moreover, the research contribution in this paper must be made clear as no new algorithm is developed but the improved or modified algorithm is proposed, i.e. incremental contribution but not a fundamental contribution.
Author response: Thank you very much for this comment.
Author action: The contribution explanation is updated.
Reviewer#2, Concern # 4: What is the difference between scheduling in cloud computing and fog?
Author response: I appreciate this comment.
Author action: One paragraph has been added to the introduction about the difference between scheduling in cloud computing and fog.
Reviewer#2, Concern # 5: The literature reviews are not properly done in this paper. It’s also important if the author can summarize the advantages, and disadvantages of each strategy in a table. Authors have missed some state of the art schedule mechanisms that would add to the novelty aspect of the paper. To that authors are suggested to compare papers such as
An improved pathfinder algorithm using opposition-based learning for tasks scheduling in cloud environment
Hybrid heuristic algorithm for cost-efficient QoS aware task scheduling in fog–cloud environment
Multi-objective scheduling technique based on hybrid hitchcock bird algorithm and fuzzy signature in cloud computing
Combining neural network-based method with heuristic policy for optimal task scheduling in hierarchical edge cloud
A two-stage scheduler based on New Caledonian Crow Learning Algorithm and reinforcement learning strategy for cloud environment
Opposition-based learning inspired particle swarm optimization (OPSO) scheme for task scheduling problem in cloud computing
Author response: Thank you very much for this comment and related works.
Author action: All mentioned related works have been added to the paper and Table 1.
Reviewer#2, Concern # 6: The reference of each equation should be written right above it.
Author response: Thanks a lot for this comment.
Author action: The reference of all equations has been added at the first explanation of them.
Reviewer#2, Concern # 7: In Section 3.3, fitness function is very simple. Why?
Author response: Thank you for a good question.
Author action: The bandwidth and the utilization are two important parameters that be considered in the fitness function. Also, due to presenting a low-level heuristic algorithm, different parts of the proposed approach are simple to implement and execute in different kinds of devices such as IoT, fog, and edge.
Reviewer#2, Concern # 8: Section 3.5 should be explained in detail.
Author response: Thanks a lot for this comment.
Author action: A pseudocode has been added for more clarification.
Reviewer#2, Concern # 9: In Section 4, the performance evaluation of the proposed method is too simple, and it lacks the comparison with other advanced algorithms proposed in recent three years.
Author response: Thank you indeed for this comment.
Author action: The proposed approach is compared with some benchmarks. Also, some other comparisons have been added according to the number of users. More extensions and comparisons with state-of-the-art research are planned for future work.
Reviewer#2, Concern # 10: How set Table1?
Author response: I appreciate this comment.
Author action: All set values for this table are based on a benchmark application as DSNS in the iFogsim library that has been used in this work.
Reviewer#2, Concern # 11: The experiment is not comprehensive and thorough. For example, what is the overhead of the proposed algorithms? Are the algorithms still effective as the scale of data increases?
Author response: Thanks a lot for the comment.
Author action: An analysis of the number of users has been added. This evaluation is based on energy consumption, total execution cost, and application delay.
Reviewer#2, Concern # 12: It is important to make it clear how other benchmarking methods are selected/employed when making comparisons against the proposed algorithm. More details are expected, otherwise, the comparison results may not be convincing. If benchmarking methods are not fine-tuned against the current problem, it means that these methods cannot perform at their very best, hence, the comparison is biased, i.e., in favor of the proposed method.
Author response: I appreciate the comment.
Author action: Since, the proposed approach is based on a hyper-heuristic algorithm, thus, a comparison with heuristic, meta-heuristic, and also multi-objective methods has been done. To evaluate a fair comparison, all device’s configurations are the same.
Reviewer#2, Concern # 13: The reason for the strong and weak performance of the methods should be explained (more discussion) in the results section (Section 4).
Author response: Thanks a lot for this comment.
Author action: Some more explanations have been added to the mentioned section.

Round 2
Reviewer 1 Report
I consider that the paper looks better now.
I am satisfied with the changes made to the initial version.
Reviewer 2 Report
The authors have made revisions according to the comments of the reviewers.